# Modified Hemagglutination Tests for COVID-19 Serology in Resource-Poor Settings: Ready for Prime-Time?

**DOI:** 10.3390/vaccines10030406

**Published:** 2022-03-08

**Authors:** Daniele Focosi, Massimo Franchini, Fabrizio Maggi

**Affiliations:** 1North-Western Tuscany Blood Bank, Pisa University Hospital, 56124 Pisa, Italy; 2Division of Transfusion Medicine, Carlo Poma Hospital, 46100 Mantua, Italy; massimo.franchini@asst-mantova.it; 3Department of Medicine and Surgery, University of Insubria, 21100 Varese, Italy; fabrizio.maggi63@gmail.com

**Keywords:** COVID-19, SARS-CoV-2, hemagglutination tests, Spike, serology

## Abstract

During the ongoing COVID-19 pandemic, serology has suffered several manufacturing and budget bottlenecks. Kode technology exposes exogenous antigens on the surface of cells; in the case of red blood cells, modified cells are called kodecytes, making antibody–antigen reactions detectable by the old-fashioned hemagglutination test. In this commentary, we review evidence supporting the utility of SARS-CoV-2 Spike kodecytes for clinical diagnostic purposes and serosurveys in resource-poor settings.

The transfusion medicine community has been involved in COVID-19 research under many different facets, ranging from blood donation shortages and COVID-19 convalescent plasma (CCP) donor identification [1] to the provision of blood components and derivatives to COVID-19 patients [2] and identifying ABO proteins as risk factors for COVID-19 severity [3]. Soon, the community could also have a role in viral serological testing.

Serological surveys during a pandemic have relevance for clinical diagnosis, epidemiology (serosurveys), convalescent plasma donation, and vaccine response monitoring. Large-scale deployment is not affordable for underdeveloped economies because of the inherent cost of dedicated kits, liquid handling, and reader devices, as well as a shortage of trained personnel. Furthermore, during the COVID-19 pandemic, a massive shortage of serology reagents persisted for months due to manufacturing bottlenecks. The majority of serological point-of-care tests (POCT) developed to date for SARS-CoV-2 rely on lateral flow immunochromatographic assays (LFIA): they are cheap and scalable, but they do not offer quantitative information or leave any record of the reaction, and they suffer from poor sensitivity. Cheap manufacturing of recombinant SARS-CoV-2 antigens (either Spike protein or peptides) has been achieved [4], but it clearly is not enough *per se*.

Blood components are within the WHO list of essential medicines, and hemagglutination tests (HAT) (based on the principle of visible red blood cell (RBC) agglutination) are pillars of modern transfusion medicine. The HAT was developed by Hirst, McClelland, and Hare in 1941 [5,6]; then, it was modified by Jonas Salk in 1944–1948 [7]. The IgM antibodies are large (35 nm) and multivalent and they are hence frequently referred to as direct hemagglutinins. On the contrary, most IgG antibodies are smaller (14 nm), unable to induce visible hemagglutination without the assistance of secondary enhancing reagents (i.e., Coombs’ serum), and commonly referred to as indirect agglutinins. The agglutinated lattice can be viewed by the naked eye and it maintains the RBCs in a suspended distribution: the formation of the lattice depends on the concentrations of both the pathogen and the RBCs, and when the pathogen or antibody concentration is too low, the RBCs precipitate at the bottom of the well.

On the other hand, the hemagglutinin inhibition (HAI) assay is performed by first mixing the pathogen with serial serum dilutions: the antibody is allowed time to bind the pathogen and then RBCs are added to the mix. Pathogens that have bound to the antibody will be unable to bind to RBCs; thus, the absence of hemagglutination is reported as a positive result for the presence of specific antibodies. If a hemagglutinating pathogen is the known reagent, the HAI assay can be used to detect the antibody. If the hemagglutinating pathogen is unknown, it can be identified by using a panel of antibodies with known specificities.

The RBCs used in the HAT and HAI assays are typically sourced from chickens, turkeys, horses, guinea pigs, or humans according to the selectivity of the targeted virus or bacterium to receptors on the RBCs; e.g., the human influenza virus HA preferentially binds to sialic acid receptors containing α2,6-Gal; avian influenza viruses preferentially bind to those containing α2,3-Gal (such as horse or chicken RBCs [8]); and *Treponema pallidum* best binds to sheep RBCs. The density and the accessibility of antigens are other key factors for lattice formation, and receptor-destroying enzymes (RDE) are commonly needed to treat samples prior to analysis to prevent non-specific binding [9]. Despite all these limitations, the high reproducibility of HAT and HAI results between laboratories has been achievable with standardization. Historical applications of HAT and HAI in virology has been limited to adenoviruses [10], measles [11], and influenza viruses [12]. As previously mentioned, HAT is not an identification assay, as many different agents have hemagglutinating properties. Additionally, live as well as inactivated viruses are both detected by the HAT. Dosage is also relevant, so that amplification by virus isolation in cell culture or embryonated chicken eggs is often required before hemagglutinating activity can be detected by a HAT. The HAI is still widely used in microbiology laboratories, but gene sequencing technology is nowadays replacing the HAI for subtype identification. Both HAT and HAI require pathogen manipulation: although attenuated strains work equally well, this still requires biosafety cabinets, trained personnel, and culture media reagents. For pandemic responses, proteins are far simpler to use as a pathogen surrogate; however, bioconjugation reactions involving whole proteins are less efficient due to steric hindrance and charge, requiring customization of reaction conditions so peptide bioconjugates are usually favored.

The HAT can be run either manually (tube technique) or on automated reading devices within transfusion services—which are available at almost any hospital worldwide—providing a quantitative score: 1 hemagglutinating unit (HAU) equals to approximately 5–6 logs of influenza virus [13]. Column agglutination technology (CAT) based on gel card cassettes facilitates reading of hemagglutination reactions; it has short reaction development times and low cost; and, it can hence be applied as a POCT.

Kode™ Technology (Kode Biotech Limited (KBL), Auckland, New Zealand) is a range of “biological paints” to create designer biological surfaces. Since the 2000s, the technology has been used to add recombinant antigens to an RBC surface, with the final product named “kodecytes”. Originally employed to create quality controls for rare antigens in immunohematology, it was soon exploited to add antigens from pathogens and aid in microbiology testing. An NIH-driven group has hence developed SARS-CoV-2 RBD kodecytes (“C19-kodecytes”) [14]. In brief, the Kode function–spacer–lipid (FSL) constructs FSL-1147 and FSL-1255, consisting each of a 15-amino acid long non-glycosylated peptide from the SARS-CoV-2 Spike protein S2 subunit (outside the receptor binding domain (RBD)). The constructs are first dispersed in red cell stabilizer solution at concentrations of 1.5 µmol/L and 2.5 µmol/L, respectively, and the blend is then incubated with washed packed group O RBCs for 2 h at 37 °C, then adjusted to 1% using an RBC stabilizer solution. Such “C19-kodecytes” are hence agglutinated in the presence of SARS-CoV-2 antibodies in a patient’s serum or plasma using routine immunohematology platforms. The reagent can be lyophilized for ease of shipping aliquots. The authors scaled up production of this reagent up to 1 g, which is sufficient for 10 million tests at a cost of less than 0.10 € per test well. When testing >120 expected negative blood donor samples and >140 CCP samples, specificity in 3 different CAT platforms (Bio-Rad, Grifols, and Ortho Clinical Diagnostics) against C19-kodecytes was >91%. Sensitivity (positive reaction rate against expected positive convalescent, PCR-confirmed samples) ranged from 82% to 97% compared to 77% with the anti-N Abbott Architect SARS-CoV-2 IgG assay. Manual tube serology was less sensitive than CAT [14]. In an external validation run in Germany and in New Zealand, Weinstock et al. investigated 130 samples from COVID-19 convalescent plasma donors using a manual C19-kodecyte assay, 2 FDA authorized ELISA assays (targeting Spike or nuclecapsid proteins), and a virus neutralization test (VNT). The sensitivity of the C19-kodecyte assay was 88%, comparable to the anti-Spike and anti-nucleocapsid ELISAs (86% and 83%) and the VNT (88%). The specificity of the C19-kodecyte assay was 90% (anti-Spike 100% and anti-nucleocapsid 97%). They further passed the assay on the Erytra^®^ Automated System (Grifols S.A., Barcelona, Spain), which uses column agglutination cards with 8 reaction columns and is capable of routinely processing at least 50 cards per hour, where they tested sera from 231 subjects receiving anti-Spike vaccines. If just screening for SARS-CoV-2-antibody, a test was to be done with an autocontrol (i.e., 4 samples per card), then 200 samples could be tested per hour, especially when the analyzer is otherwise idle. However, as a positive autocontrol result (being unmodified cells used to make kodecytes) is due to the presence of natural red cell antibodies, the autocontrol need not be done when testing blood donor populations who are already screened for red cell antibodies. Therefore, when screening blood donors 8 samples could be tested per column card, allowing for potentially 400 tests per hour [15]. The same NIH group evaluated 140 convalescent COVID-19 patients and 275 healthy controls using the C19-kodecyte assay. The analytical performance of the new assay was compared with a VNT and two commercial chemiluminescent antibody tests (Total assay and IgG assay, Ortho). The C19-kodecyte assay detected SARS-CoV-2 antibodies with a sensitivity of 92.8% and a specificity of 96.3%, well within the minimum performance range required by FDA for EUA authorization of serologic tests. The Cohen’s kappa coefficient was 0.90 indicating an almost perfect agreement with the total assay. The Pearson correlation coefficient was 0.20 with the neutralizing assay (0.49 with IgG and 0.41 with total assays), which can be explained by the smaller number and different Spike epitope and their masking by carbohydrates and proteins on the glycocalyx of RBCs [16].

Kodecytes are not the lone exploit of the HAT for COVID-19 serology, since antigens can alternatively be bound to RBCs via nanobodies targeting ubiquitous RBC antigens (such as glycophorin A and H antigen) (Figure 1). This approach has been pursued in the past using anti-glycophorin A mAbs such as 1C3/86 and 10F7MN [17], which were successfully bioconjugated to HIV gp41 during the early years of the AIDS pandemic [18,19], and it has hence been successfully implemented for SARS-CoV-2.

Townsend et al. in Oxford used the single domain antibody (nanobody) IH4, targeting a conserved glycophorin A epitope on RBCs (previously used to coat RBCs with HIV p24 [20]), linked to the RBD from SARS-CoV-2 (amino acids 340–538) via a short (GSG)2 linker to produce the fusion protein IH4-RBD-6H. Such a reagent can be lyophilized for ease of shipping. Such a HAT test had a sensitivity of 90% and a specificity of 99% for detection of antibodies after a PCR diagnosed infection. The HAT could be titrated, detect rising titers in the first five days of hospital admission, and it correlated well with the Siemens Atellica Chemiluminescence assay for detection of IgG antibodies to the RBD. Production of this reagent was scaled up to 1 g, which is sufficient for 10 million tests at a cost of ~0.27 UK pence per test well [21]. The assay was later correlated by Nguyen et al. with microneutralization using authentic and pseudotype VNT specific for SARS-CoV-2 variants of concern Alpha and Beta [22].

Kruse et al. at Johns Hopkins University, using a fusion of the SARS-CoV-2 Spike RBD to the single-chain variable fragment (scFv) 2E8, targeted the H antigen on RBCs previously used to attach HIV gp41 [23]. A total of 200 COVID-19 patients and 200 control plasma samples were reconstituted with O-negative RBCs to form whole blood and added to the dried protein, followed by a stirring step, a tilting step, and a 3-min incubation, and then a second tilting step. The sensitivity for the HAT, Euroimmun IgG ELISA test and RBD-based CoronaCheck lateral flow assay was 87.0%, 86.5%, and 84.5%, respectively, using samples obtained from recovered COVID-19 individuals. Testing pre-pandemic samples, the HAT had a specificity of 95.5%, compared to 97.3% and 98.9% for the ELISA and CoronaChek, respectively. A distribution of HAT strengths was observed in COVID-19 convalescent plasma samples, with the highest HAT score (4) exhibiting significantly higher neutralizing antibody (nAb) titers than weak positives (2) (*p* < 0.0001). Strong agglutinations were observed within 1 min of testing, and this shorter assay time also increased specificity to 98.5% [23].

Alves et al. combined CAT with bioconjugates consisting of three different Spike peptides with anti-Rh(D) antibodies, which facilitate RBC cross-linking only in the presence of plasma containing antibodies against SARS-CoV-2 [24].

Redecke et al. developed NanoSpot.ai, a modified HAT that employed a dromedary-derived bispecific nanobody (H11-D4) binding RBCs and Spike RBD, and a smartphone tool for digital image acquisition and interpretation. The assay correlated well with both the Euroimmun ELISA and the Abbott CLIA assays [25].

Esmail et al. coated group O Rh(D)-positive RBCs with the Spike RBD or the RNA-binding domain of the nucleocapsid (N-RBD) protein through streptavidin–biotin mediated coupling and showed a correlation with an ACE2-binding surrogate VNT [26].

The platforms discussed above clearly differ in the Spike epitopes used (RBD vs. non-RBD). The C-19 kodecytes, targeting a Spike region outside the RBD, have the theoretical potential to discriminate convalescents from Spike vaccinees. Not by chance, in the study by Weinstock et al., 34 out of 73 vaccinees were seronegative after vaccination and only 8 became positive over time, when epitope spreading likely expanded the repertoire of vaccine-elicited antibody specificities [15]. Further validation is nevertheless required before this indication can be confirmed.

Much also remains to be learned about SARS-CoV-2 infection of RBCs: SARS-CoV-2 attaches to sialic acid sites RBCs in vitro [27], although in vivo SARS-CoV-2 has never been cultured from blood [28]. While seasonal (endemic) betacoronaviruses express hemagglutinin esterase (HE) to facilitate virion release and budding, the virulent (SARS-CoV-2, SARS-CoV-1 and MERS) do not, facilitating clots in pulmonary and systemic microvasculature [29].

Overall, diagnostics for the developing world should meet the ASSURED (affordable, sensitive, specific, user-friendly, rapid/robust, equipment-free, deliverable to end-users) criteria [30]. Much remains to be done in terms of validation to create confidence in modified HAT diagnostics (e.g., the influence of hemoconcentration in hospitalized patients or cross-reactivity with endemic coronaviruses), but they should be considered as frontline diagnostics for low-resources settings in future pandemics.

## Figures and Tables

**Figure 1 vaccines-10-00406-f001:**
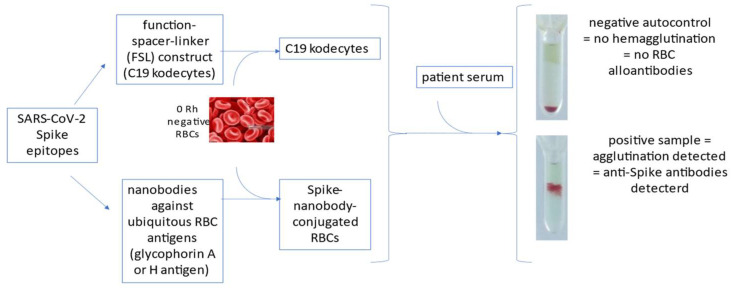
Summary of HAT-exploiting technologies for COVID-19 serology. Anti-Spike antibodies can be detected by hemagglutination when epitopes are bound to red blood cells (RBCs) either via function–spacer–linker (FSL) constructs or nanobodies.

## Data Availability

The data presented in this study are openly available in PubMed, bioRxiv and medRxiv.

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
