# Peer review of "Modified Hemagglutination Tests for COVID-19 Serology in Resource-Poor Settings: Ready for Prime-Time?"

_vaccines, 2022, doi:10.3390/vaccines10030406_

Round 1

Reviewer 1 Report

Proposed corrections. The caption for Figure 1 should be more complete.

The Best

Author Response

Caption for Figure 1 has been expanded as requested to be clearer

Reviewer 2 Report

In this manuscript, the authors reviewed evidences supporting the utility of SARS-CoV-2 Spike kodecytes for clinical diagnostic purposes and serosurveys in resource-poor settings. They summarized the HAT (hemagglutination tests)-exploiting technologies for COVID19 serology. The authors concluded that although much remains to be done in terms of validation to create confidence in modified HAT diagnostics, they should be considered as frontline diagnostics for low-resources settings in future pandemics. This work is interesting and informative. Therefore, I recommend the paper to be accepted by Vaccines after addressing the following issues in a minor revision.

  1. What does HAI abbreviate for? The authors should explain it in the paper.
  2. The authors mentioned several examples about C19-kodecyte assay. It would be better if the authors could include a table or a chart describing the sensitivity and specificity instead of barely include the numbers in the text.
  3. The authors should double check the format of references. For example: ref. 21, 26.

Author Response

We have spelled out the HAI acronym at first occurrence as requested.

We feel the limited number of references to C19-kodecytes and their heterogeneity discourages creating a Table, and prefer to leave them under text.

References are often ePub ahead of print (missing volume and page ranges), and will be fixed at proof stage

Reviewer 3 Report

This is a potentially interesting manuscript which should be made more understandable for a briter readership.

1.Please add some more information to the abstract

2.transfusion medicine is also involved in providing albumin for hospitalized COVID-19 patients (Ramadori G,Int J Molc Sci 2021.albumin infusion in severely ill COVID-19 patients.

3.hemoconcentration should be considered when testin hospitalized patients and

4.how about pre-existin coronavirus antibodies,which serum dilution is necessary to avoid positive test results.

5.Line 178 the sentence gives the impression that attachment of the viru to RBC could happen in vivo and may suggest that COVID-19 also enters systemic circulation.This has not been shown and virus from the blood circulation has not been cultured so far

Author Response

We have expanded the abstract.

We have added blood provision of blood components and derivatives to COVID19 patients to the list of roles transfusion physicians have in COVID19. 

Since no data are available yet on the effect of hemoconcentration or cross-reactivity with endemic coronaviruses, we have added them as caveat in the bottom paragraph.

When discussing SARS-CoV-2 infection of red blood cells, we have specified this has been shown in vitro, while in vivo the virus has not been cultured from blood yet.

Round 2

Reviewer 3 Report

Authors should introduce references:

line21:mention the paper Violi et al.Thromb.Hemost 2021;12:102-105

line 182:please mention references for in vitro (infection of red blood cells) and in vivo studies (no virus can be cultured from blood samples).

Author Response

The suggested references have been added.